# ‘Misdiagnosed and Misunderstood’: Insights into Rarer Forms of Dementia through a Stepwise Approach to Co-Constructed Research Poetry

**DOI:** 10.3390/healthcare12040485

**Published:** 2024-02-17

**Authors:** Paul M. Camic, Mary Pat Sullivan, Emma Harding, Martha Gould, Lawrence Wilson, Sam Rossi-Harries, Adetola Grillo, Roberta McKee-Jackson, Susan M. Cox, Joshua Stott, Emilie V. Brotherhood, Gill Windle, Sebastian J. Crutch

**Affiliations:** 1Dementia Research Centre, Department of Neurodegenerative Disease, UCL Queen Square Institute of Neurology, University College London, London WC1N 3AR, UK; emma.harding@ucl.ac.uk (E.H.); s.rossi-harries@ucl.ac.uk (S.R.-H.); r.mckee-jackson@ucl.ac.uk (R.M.-J.); e.brotherhood@ucl.ac.uk (E.V.B.); s.crutch@ucl.ac.uk (S.J.C.); 2School of Social Work, Faculty of Education and Professional Studies, Nipissing University, North Bay, ON P1B 8L7, Canada; marypat.sullivan@nipissingu.ca (M.P.S.); a.tola@outlook.com (A.G.); 3Independent Researcher, North Bay, ON P1B 8L7, Canada; 4Independent Researcher, Sonic Studios, Rye, East Sussex TN31 7NY, UK; wilsonlawrenceuk@gmail.com; 5W. Maurice Young Centre for Applied Ethics, School of Population and Public Health, University of British Columbia, Vancouver, BC V6T 1Z3, Canada; susan.cox@ubc.ca; 6Research Department of Clinical, Educational and Health Psychology, University College London, London WC1E 6BT, UK; j.stott@ucl.ac.uk; 7Ageing and Dementia @ Bangor, Dementia Services Development Centre, School of Health Sciences, Bangor University, Bangor LL57 2EE, UK; g.windle@bangor.ac.uk

**Keywords:** non-memory led dementia, young onset dementia, inherited dementia, care-partners (care-partner and carer are used interchangeably in this paper), carers, caregivers, healthcare professionals, virtual environments, qualitative, arts and health

## Abstract

This study investigated co-constructed research poetry as a way to understand the lived experiences of people affected by rarer dementia and as a means to use poetry to convey those experiences to healthcare professionals. Using mixed methods, 71 people living with rarer dementia and care-partners (stakeholders) contributed to co-constructing 27 poems with professional poets; stakeholders’ verbatim words were analysed with descriptive qualitative analysis. Stakeholders were also surveyed and interviewed about their participation. Healthcare professionals (*n* = 93) were surveyed to elicit their responses to learning through poetry and its acceptability as a learning tool. Poems conveyed a shared narrative of different aspects of lived experience, often owing to atypical symptoms, misunderstandings by professionals, lack of support pathways, and a continuous struggle to adapt. Stakeholder surveys indicated it was a valuable experience to both co-create and respond to the poems, whilst group interviews revealed people’s experiences of the research poetry were characterised by reflection on lived experience, curiosity and exploration. Healthcare professionals’ responses reinforced poetry’s capacity to stimulate cognitive and affective learning specific to rare dementia support and prompt both empathy and critical thinking in practice. As the largest poetry-based study that we are aware of, this novel accessible approach of creating group poems yielded substantial information about the experiences and needs of those affected by rarer dementia and how poetry can contribute to healthcare education and training.

## 1. Introduction

Differing from more prevalent, mainly older onset Alzheimer’s disease, there are rarer forms of dementia characterised by non-memory-led symptoms, such as vision and spatial skills (posterior cortical atrophy), language and communication (primary progressive aphasia), behaviour and social interactions (frontotemporal dementia), autosomal dominant inheritance patterns (familial Alzheimer’s disease and familial frontotemporal dementia) or hallucinations, fluctuations and movement (Lewy Body dementia). Rarer dementia disproportionately affect those under 65 (young onset dementia), bringing a range of difficulties and issues associated with being of working age and dependent on employment, caring for children at home and remaining too young for state benefits. An overall international age-standardised prevalence of 119.0 per 100,000 population in the 30–64 age range was more recently estimated for younger onset dementia [1], equating to 3.9 million people worldwide. Hendriks et al. [2] also estimated an overall age-standardised young onset dementia (YOD) incidence rate of 11 per 100,000, corresponding to 370,000 new cases annually worldwide. 

The experience of obtaining an accurate diagnosis and relevant care has often proved to be an arduous process, especially for those from underrepresented communities [3], negatively affecting the lives of people living with rarer forms of dementia (PLwRD), including family members and other informal caregivers [4]. Because of differences in symptoms and age of onset, providing support through the same programming (e.g., Memory Cafés) as those with typical Alzheimer’s or vascular dementia has proven problematic. In addition, because of their rarity, these types of dementia are less well recognised by healthcare professionals, health organisations, and dementia charities, and still continue to provide diagnostic challenges over a decade after Rossor et al. [5] brought them to the forefront of clinical practice. 

Across millennia, poetry has conveyed sensory and emotional information, offered a way to understand and appreciate complex and lesser-known phenomena, and acted as a conduit for entertainment and engagement. Poetry can be personal and subjective, descriptive and accessible, polysemantic, difficult to comprehend, and embedded with hidden meaning [6]. Poetry also offers different possibilities for participant-voiced and researcher-voiced poems [7,8] and can also generate emotional and sensory experiences, as well as knowledge, “and thus help bridge the gap between ourselves and the ultimately impenetrable experience of dementia” [9] (p. 162). In a research context, poetic inquiry can be used as part of an analytic device in data collection and dissemination [10], as a methodological tool to interpret lived experience and generate knowledge in unique ways to bring new insights and to present findings to peers and the general public [11] (p. 88). As an art activity used as an intervention and target activity, poetry has the flexibility to be read, listened to, and created in both individual and group formats [12], making it an accessible activity for people with different types of dementia and at different levels of impairment [13,14]. It can also be a co-created participatory activity involving PLwRD, care-partners and experienced poets [15]. 

Within a dementia population, poetry offers the flexibility of being accessible in different settings [16] including one’s own home, community venues, day and residential care, and can involve family, friends and care workers [17]. It is also inclusive because it challenges positivist bio-medical orthodoxies around care and gives space for people whose experience is not mainstream, offering a new set of tools to enable them to express their experiences in a different way [18]. Hurren puts it succinctly when she explains what poetry offers: “…not a form of abstract, quantifiable knowledge…that can be amassed and advanced. Instead, through the convenient portability of words, it offers a semblance of direct experience, a recovery or approximation of emotional experience that engages our sense of the numinous and aesthetic” [19] (p. 229). The typical uses of poetry as an artistic activity or intervention have sometimes relied on the spoken word, like traditional forms of qualitative research such as interviews, which may unwittingly exclude persons with symptoms such as aphasia, common in some rarer forms of dementia, and people with more advanced stages of dementia. Likewise, sole reliance on extensive writing may exclude people with dysgraphia [20]. As rare dementia is not well understood or widely researched, research poetry has the potential to help communicate to healthcare professionals and students the emotional and social lives of people living with different neurodegenerative diseases. 

### Research Aims

The aims of this study investigated co-constructed research poetry as a way to understand the lived experiences of people affected by rarer dementia, and as a means to use poetry to convey those experiences to healthcare professionals. Through the use of language, form and structure, poetry pushes us to look at things in different ways than a lecture, textbook or diagnostic manual. Poetry is an emotive and evocative artform that asks questions, can be destabilising and disruptive, and is a powerful medium for diverse voices to speak, and for other people to listen to those voices. In order to allow any reader, regardless of background, “an immersive glimpse into the emotional world” [21] (p. 2) of this population, though discussion with stakeholders (PLwRD and family carer-partner participants), we sought to collaboratively create research-informed poems based on the diverse lived experiences of PLwRD. Through a stepwise approach [22], our main aim was to use research-informed poetry to convey affective, social and experiential aspects of lived experience of rarer forms of dementia to healthcare professionals. Before pursuing this aim, however, we first needed to know from our stakeholder writers—the first audience of readers—their reactions to the poems; broadly desiring to know if the poems had validity for *them*; whether/how the poems translated *their* lived experiences (e.g., were they meaningful, informative, worthwhile, trustworthy, helpful or useful ways to communicate these experiences).

## 2. Materials and Methods

### 2.1. Study Design 

This was a mixed-methods study primarily involving qualitative data from co-constructed poems and survey responses from PLwRD and care-partners, along with group interviews of care-partners. Healthcare professionals were subsequently surveyed about their responses to listening to/reading a selection of poems. 

### 2.2. Participants and Recruitment

Stakeholders-invited (*n* = 102) and eventual participants (*n* = 71) included PLwRD (N = 27, 49–82 years) and care-partners (*n* = 44, 36–78 years). The study involved people affected by familial Alzheimer’s disease (FAD), behavioural variant frontotemporal dementia (bvFTD), primary progressive aphasia (PPA), posterior cortical atrophy (PCA), Lewy body dementia (LBD), young-onset vascular dementia (YOVaD), and young-onset Alzheimer’s disease (YOAD) (Table 1). Characteristics of the dementia types included in the present study (found in Supplemental Appendix A [5,23,24,25,26,27,28]) represent a range of symptom profiles and support needs. Recruitment involved emailing information about this project to stakeholders who had previously consented to other research activities within a larger international study on rare dementia support [29]. Stakeholders were recruited via email from (name of rare dementia organization, removed for anonymous review). Healthcare professionals (*n* = 93, Table 2) were recruited through different professional networks. The study was approved by (blinded) ethics committee in the United Kingdom and (blinded) ethics committee in Canada. We made a pragmatic decision to include care-partners and people living with dementia because the experience of relations with others is a key theoretical and practical component in dementia care [30,31]. For PLwRD and care-partners, there were no general exclusion criteria nor were there exclusions based on current medications at the time of consent. Inclusion criteria: minimum age of 18 years old; can understand and speak English or Welsh; has the capacity to consent independently; resides in the community; and has access to the internet by computer, tablet or smartphone and an email address. For healthcare professionals there were no exclusion criteria and for student-trainees, they had to be currently enrolled in a university training programme.

### 2.3. Data Collection 

Our stepwise approach to data collection (Figure 1) involved five components (described below) and utilised maximum variation sampling, a type of purposeful sampling, in order to obtain “broad insight” [32] (p. 2) from those affected by rarer dementia.

(1) PLwRD and care-partners were invited to contribute their own words (source material) to the development of 3 poems over a 12-week period that highlighted different aspects of their lived experiences. The contributions of PLwRD and care-partners were combined in the same poems. Neither PLwRD or their care-partners exist in isolation. The importance of including both PLwRD and their care-partners was paramount, as their lives are linked together and they are experiencing shared life changes, making dementia a shared experience [33,34]. Colleagues from our charity partner (blinded) and lived experience consultants (people who have personal experience of rarer forms of dementia as a PLwRD or as a care-partner) felt it was important for the voices of PLwRD and care-partners to be conjointly heard. An additional motivation for combining PLwRD and care-partner contributions was to counter an otherwise potential biassing effect related to the nature of cognitive impairments experienced by different diagnostic groups (e.g., experiences related to PPA might be under-represented in any PLwRD-only exercise owing to challenges with language production and comprehension). In addition, the study’s exploratory design was not seeking to compare experiences between PLwRD and care-partners. After the project was explained and consent had been taken, participants were emailed a series of 3 prompts about 4 weeks apart. Each prompt began with, Using from one word up to a sentence (and continued with one of the following): What is your experience of having (type of dementia) or being a family member caring for someone with this type of dementia?How would you describe (type of dementia) to a friend or family member?What does support mean to you?

The prompts were oriented toward a specific type of rarer dementia (PCA, FTD, PPA) involving people from England, Singapore, Switzerland, Wales or sent to people in mixed-rare dementia (MRD) cohorts (PCA, FTD, PPA, YOAD, LBD and YOVaD) from Canada; there were nine cohorts ranging from 4 to 13 people, determined by rolling recruitment over the course of the study. Cohort size was flexible and was determined by recruitment. Responses were emailed back to a research team member and anonymised before being sent to one of our facilitating poets (MG or LW) who created a poem within about 2 weeks. Unlike many forms of research poetry, the facilitating poets did not bring their own words into a poem but wrote exclusively using participants’ words except for linking words. About eighty percent of the source material was used for the poems, ensuring that contributions from everyone were included. The completed poems were also audio-recorded.

(2) After each of 3 poems were completed, written and audio-recorded versions of the poems were returned via email to participants, along with a description of the facilitating poet’s creative process, the original source words, and a link to an anonymous online survey using Qualtrics^®^ software (version 1/2022) [35], asking the following questions: 

(a) What was your experience of writing the words that you did? 

(b) What did you think and feel about the completed poem? Please be candid, we want to know your opinion of the poem and your thoughts about the process of taking your words and making them into a poem. 

(c) Prior to creating this poem have you written poetry, read poetry or attended a poetry reading within the last 5 years? [included only after the first poem].

(3) Four group interviews were conducted with 9 care-partners and 1 PLwRD (no additional PLwRD volunteered) to obtain more in-depth information about their experiences of participating. 

(4) Individual interviews occurred with both poets about their creative process and experience participating in co-created poetry (reported in a separate paper). 

(5) The final part of the study involved healthcare professionals and students-in-training responding to up to 3 poems in an anonymous, mixed-methods online survey, also using Qualtrics^®^ software [35]. The 20 minute survey invited participants to consider poetry as a learning tool by reading and listening to three poems with follow-up questions after each poem: 

(a) Describe your immediate reaction after reading and listening to the poem. Consider your emotions, senses, affect, thoughts, memories, sounds. If it helps to communicate your responses to this poem; please tell us if there is a particular word(s) or phrase(s) that you are responding to. 

(b) In regards to providing support and care, what story/narrative is this poem describing to you as current or future health/social care professional; 

(c) What did you discover about people living with rare or young onset forms of dementia and/or their care-partners from this poem? This was followed by demographic questions (e.g., profession, work or personal experience with someone with dementia) and involvement in reading or writing poetry over the last 5 years. A brief explanation of the research poetry process was also provided. 

### 2.4. Data Analysis 

The co-constructed poems and survey responses from stakeholders were analysed using a qualitative descriptive (QD) approach to thematic analysis, described by Sandelowski [36,37], and informed by the work of Kahlke [38] and Kennedy [39]. QD has been shown to be highly relevant for health services and policy research [40]. The present study was guided by wanting “to provide a rich, straight description of experience…in a language similar to the informants’ own language” [32] (p. 2) rather than adhering to theoretical constraints or developing higher level interpretive conceptualisations, which, whilst valuable, were not related to our aims of wanting to attend to as closely as possible to meanings of experiences, as described by participants [41]. Healthcare professionals’ survey data were also analysed using a QD thematic analysis [36,37].

Data analysis consisted of 4 steps (Figure 2) and was influenced, in part, by Macdonald et al. [42].

1. Each of 4 data units (from stakeholders: poems, surveys, group interviews and from healthcare professionals: surveys), seen in Figure 1, were first inductively and independently open coded and a codebook was developed to guide further coding. 

2. Using an intercoder consensus approach [43], further analytic discussions resulted in the development of themes. 

3. Guided by the pioneering work of Sandelowski [44], narrative synthesis [42] was used to bring together the themes from each data unit to ensure different perspectives were represented. The synthesis sought to reduce data fragmentation and cohesively bring separate accounts together. 

4. The final step, cross-synthesis analysis and interpretation, sought to bring out key findings, looking for what was unique and discordant [42], and what might be missing.

## 3. Results

### 3.1. The Poems

Facilitating poets received words (source material) from 71 stakeholders and co-constructed 27 poems. Results are organised by prompts and themes together with three representative poems below (representative poems were chosen to convey different forms of rarer dementias across the three prompting questions). Table 3 contains a list of prompts, themes and titles of all 27 poems. Table 4 contains themes, sub-themes and illustrative quotes organised by prompt. The entire collection of poems, accompanied by the facilitating poets’ explanations for each poem, are in Appendix A. 

#### 3.1.1. Prompt 1 (for Poems 1–9, Appendix A): What Is Your Experience of Having (Type of Dementia) or Being a Family Member Caring for Someone with This Type of Dementia? 


**Out Here in the Rain**



*“We are not the walking dead”*


I’m stuck here with a rare disea … eeeze

  *devastating*


  *terrifying*

  *heart breaking*

  *frustrating*

The physicians don’t have a clue, who? 

  *very maddening*


  *very stressful*

  *very sad*

I’d be stuck in last century

sitting here clueless

  *fear*

  *stigma*

You blessed me with a nursing degree

  *opportunity*

  *education*

  *support*

  *understanding*

Becoming and blossoming into something different

  *yet beautiful*


Finding a deep sense of self 

  *freeing*

  *comical*

Hours fly by like minutes

faces I ought to remember

no longer provide me names

I read and read about how to ease the pain

but none of the doctors can agree … ee

  *leaving me out here in the rain*

  *with all the pain*

The overarching theme, Temporality Disrupted (Table 4), captures the constant challenges of day-to-day life and also casts ahead at a disrupted future. Three sub-themes, *Uncharted Territory*, *Being in the Moment,* and *Re-discovering Identities and Relationships* highlight some of the enormous challenges of Temporality Disrupted, but also provide glimpses of the importance of focusing on positive moments in the present as a backdrop to an unknown, terrifying future. Temporality as a psychosocial construct has its modern roots in psychoanalytic and phenomenological theories and focuses “on subjectivity and its capacity for a temporization of events can shed light on the temporal constitution of experience, especially when this involves traumatic phenomena” [45] (p. 37). Stakeholders’ poetic contributions illuminated often hidden and private traumas that generated disruptions in identity and in family, social and professional relationships, accompanied by a loss of dignity [46]. Over time, trauma is intensified by the lack of support in their social environment of family and friends. The unfamiliarity and unexpectedness of the diagnosis was reflected in the poems, and generally associated with intense emotions, reflecting deep losses. The sub-theme *Uncharted Territory*, like much of the analysis, reflects a shared narrative of both PLwRD and care-partners’ experiences as unfamiliar and unexpected, owing to the atypical symptoms and age of onset (e.g., “letter, number and word finding difficulty” (11.12.FTD.PLwRD-1); “hallucinations” (6.1.PCA.PLwRD-1)) that were not well understood, even by health providers (e.g., “the physicians don’t have a clue” (12.5.LBD.C-1). Subjective appraisals of emotions as seen in Table 4 are not unfamiliar in the broader dementia literature [47], but take on new developmental perspectives with a different sense of temporality, in younger-onset and less well-known dementia. *Being in the Moment* was seen as a process of adapting to and accepting a new reality, rather than a one-time experience. Furthermore, it was sometimes associated with finding the lighter side of the journey. Despite many changes, people living with dementia and their care-partners also described their efforts at *Re-discovering Identities and Relationships*, which involved reflecting on where they now were in the world and reaffirming the importance of their connections with others. 

#### 3.1.2. Prompt 2 (for Poems 10–18, Appendix A): How Would You Describe (Type of Dementia) to a Friend or Family Member?


**Where to begin?**


Where to begin?

  Brain blindness

  Vision is darker. That’s not a rubber mat?

  I can see it, but at the same time I can’t see it.

  Confusing, disorientating.

  Good days and foggy days

  Frightening

Where to begin?

Which way round does this go?

  Things aren’t where they appear to be

  I wish you could see through my eyes

  Invisible dementia

  Misdiagnosed and misunderstood

  Getting confused more often

  How to describe something

  No more words

Which way round does this go?

Why can’t I find things?

  I wish you could see what I see 

  It’s a black hole 

  I can’t do anything I used to be able to do

  Eyes are okay in themselves

  No more driving

  Staying positive as possible

  How to begin?

Why can’t I find things?

Where is the top or the bottom?

  When I struggle to understand

  Word searching

  Things aren’t where they appear to be

  The brain effects the eyes

  Embuggerance

  No more words

  Misunderstood

Where is the top or the bottom?

Where is the top or the bottom? 

Why can’t I find things? 

Which way round does this go?

Where to begin?

A Rollercoaster seems to sum up the primary theme, scaffolded by the nuances found in sub-themes, *Heartbreaking Losses*, *Shifting Selves and Changing Worlds, Misdiagnosed and Misunderstood* and *I Can*. Describing their specific type of dementia to friends and family conveyed vivid but complex social and psychological disruptions to their lives filled with penetrating feelings of devastation and fear. PLwRD and care-partners both described changes in independence, abilities and symptoms as part of being on a Rollercoaster. Most conspicuously described were the intensely painful *Heartbreaking Losses* of physical and mental functions, relational interactions along with a sense of “being invisible”. *Shifting Selves* encompasses changes to their personhood [48] and relationships [46], which were also portrayed as in flux and now lacking familiar consistency. Reflected in the anger and anguish of being *Misdiagnosed and Misunderstood* included initial experiences seeking to find out what was wrong. Being of a younger age and having atypical symptoms often initially led to an inaccurate or no diagnosis, whilst also being referred multiple times to different specialties. Trying to inform sceptical family and friends who questioned and doubted whether the person actually had dementia (e.g., because they were “too young”, “their memory is fine”, and “It’s a mental health thing”) further exasperated feeling misunderstood and underlined the sense of going it alone. Yet, despite the emotional devastation accounted for in the poems, there were moments of relief and positivity. A sense of *I Can* was attained by reflecting on life prior to the diagnosis and cultivating feelings of gratitude. For others this came from living in the moment and having a sense of agency. For some, positivity was about retained abilities and qualities. 

#### 3.1.3. Prompt 3 (for Poems 19–27, Appendix A): What Does Support Mean to You?


**Help Me**


It’s a range of things --

compassion, understanding, love --

someone who listens … and believes me

someone who helps when I’m down and out

someone who reaches out

and gives me another path or crutch (an option)

someone to hug me and tell me it’s ok

someone who can take over when it’s too much

In between

there is support that helps me see things differently

and support that just takes time

to listen and see things the way I do,

not judged by expression of my thoughts or feelings

Never tell me I’m not doing enough for my husband --

just step in to help me!

Help me, so I can help him.

Honour his wishes.

Help is a range of things.

(an option: leave cookies on our porch!)

The overarching theme from this set of poems is Multidimensional Support, a complex interaction of temporal, emotional, and interpersonal components identified by sub-themes, *Not in This Alone*, *Compassion and Understanding,* and *Support as an Action*. As noted previously by stakeholders, obtaining care and support for rarer forms of dementia was often extremely challenging and time consuming. *Not In This Alone* was one of the primary consequences of support that actively countered the sense of isolation. Meeting peers who are experiencing similar life events to discuss, learn from—and also offer help to—was a cornerstone for carers. For those living with dementia, this theme encapsulated support as helping to balance out the vicissitudes of daily struggles. Not having symptoms and age of onset typically synonymous with more prevalent types of dementia, nor well understood by healthcare professionals, contributes to a sense of being alone, yet when symptoms and concerns are not dismissed, support is felt. In situations when PLwRD are able to access knowledgeable healthcare professionals and specialist charities offering diagnosis-specific peer support there is a marked sense of being less alone. Strongly echoed by many, feeling less isolated could be achieved if more healthcare professionals become knowledgeable about rarer types of dementia. 

The meaning of support, regardless of dementia diagnosis, is likely to be grounded in experiencing compassion from others [49]: a “sympathetic consciousness of others’ distress with a desire to alleviate it” [50] (p. 169). This was also reflected from stakeholders in the present study. *Compassion and Understanding* involved familiarity, dependability, and strength. Being believed was an important component of this theme for both PLwRD and care-partners given that these conditions manifest in such atypical ways, contrary to common assumptions, and in contrast to more typical memory-led dementia. Linked to *Compassion and Understanding*, *Support as an Action* for PLwRD can mean a duality of action and social activity that helps to enable, increase confidence and provide structure. The poems conveyed, from care-partners and PLwRD, that support was bi-directional, encompassing being helped but also helping others. As described by Keyes et al. [51], helping others is part of the mechanism of providing peer support for this population, which “shifts the focus away from people with dementia as passive recipients” (p. 572) of support to active deliverers.

#### 3.1.4. Poems: Narrative Synthesis 

Traumatic experiences can “generate a disruption in the person’s identity and sense of reality, which deeply involve the dimension of time” [45] (p. 37). As these poems suggest, the temporisation of experience is a shared narrative felt by both care-partners and those living with rarer forms of dementia. The poems resonate with the growing literature on young onset and rarer forms of dementia that highlight the challenges in obtaining a diagnosis and receiving appropriate care across the caregiving trajectory, the absence of awareness of atypical conditions by healthcare providers, and a dementia care system that focuses on those over age 65 [2,4]. The poems provide a range of insight into the varied experiences of both PLwRD and care-partners. This “unfolding group narrative is made up of multiple voices each shaped in relation to each other” [52] (p. 2), described by Bakhtin [53] as “A plurality of independent and unmerged voices and consciousnesses, a genuine polyphony of fully valid voices” (p. 6). Further understanding of ‘in the moment’ temporality, and what occurs before and after each moment [54], through targeted poetry prompts and discussion of completed poems, could help promote the development of better support practices co-developed with this population.

### 3.2. Stakeholder Survey Findings

This section reports on the findings from stakeholder surveys, sent to participants after they received each of three poems. As a reminder to readers, cohort members responded electronically to prompts and surveys and only ‘knew’ each other through reading the completed poems. Knowing how stakeholders made writing decisions and how they interacted with the finished poems was an essential part of our stepwise approach to developing co-constructed poetic inquiry. Knowing this information also helps us to understand what the impact of the research was on stakeholders as research participants, “an ethical question with little empirical evidence” [55] (p. 16) that continues to be vastly overlooked even in studies on sensitive personal topics [56]. 

Firstly, we report PLwRD and care-partner experiences writing (Table 5), followed by their responses reading/listening to the poems (Table 6). We asked two open-ended questions about their participation: ‘What was your experience of writing the words that you did?’ and ‘What did you think and feel about the completed poem?’. Responses were analysed using descriptive thematic analysis. Out of a possible total of 210 responses (71 participants × 3 sets of poems), 148 (70.4%) were received from care-partners (77), PLwRD (57) and bereaved carers (14). Response rates for PLwRD and care-partners were both 68 percent, whereas for bereaved carers it was 93 percent. Nearly 44 percent had participated in a poetry activity within the last 5 years.

#### 3.2.1. People Living with Rarer Forms of Dementia: Writing

Although response rates were the same, PLwRD–stakeholders provided fewer and sometimes briefer overall responses, which may be attributed to various difficulties related to their disease and/or the clarity of the survey questions. Yet there was a sense of engagement and agency reflected in the theme, *I Wanted to Do It*. The quality of responses were vivid, meaningful and candid as reflected in the sub-themes *Challenging but Enjoyable, Hurts So Much*, and *Purposeful*. Many experienced writing as enjoyable and overall, as a positive experience that they found easy to engage with. Others responded that it was also *Challenging* but across varying degrees that included enjoyment and involvement in a new “positive” activity. In *Hurts So Much*, difficulties in writing were attributed to cognitive and emotional factors related to the disease but there were also tempered concerns about getting into a dark place that might be difficult to emerge from, most noticeable from writing in response to the second prompt (Table 3). 

As with all analytic decisions, we were concerned about what might be left out and the limitations of a singular thematic category. What follows is an example of what was encountered by one participant across their contributions to two poems, and our coding of the theme we identified as Purposeful: 

“Initially, it was hard to get down on paper the points that I wanted to bring forward as there are so many aspects around living with my dementia. I did not want to overload my reply with too many aspects and responses. I was happy with my final response in the knowledge that others may express the aspects of living with the condition that I didn’t cover”(11.12.FTD.PLwRD-1).

“Initially, I needed to establish in my mind what I wanted to portray in a clear, concise way. Having hit on the idea that so often in my mind I feel jumbled and mixed up, I was then able to use the mental picture of a Tumble Drier/Washing machine as a simile for me to clearly portray how living with dementia often feels to me. I was pleased with the result” (11.12.FTD.PLwRD-2).

In the example above, the writer responding to two different prompts (Table 3) described how he made writing decisions. The dual-process model of writing [57] provides one way to understand the writing process. The model involves knowledge-retrieval (drawing on an explicit store of knowledge) and knowledge-constituting (synthesising content) approaches to writing” [57] (p. 22). Stated another way, “text production involves discovery because the original memories that are stored in episodic and autobiographical memory are not necessarily coherent or organised” [58] (p. 3), and poetic writing offers one way to synthesise and make coherent, personal experiences. Yet a word of caution from a care-partner commenting about her daughter’s inability to write: “As to being helpful for people living with rare dementia, I think it could be helpful for them, depending on their level of ability. I know that my daughter, who had BvFTD couldn’t have written anything” (9.3.FTD.C-1).

#### 3.2.2. Care-Partners: Writing

Involvement in writing was complex and varied across the three prompts, as it was for PLwRD, yet in some ways it appeared to be a Consolidating experience, with three sub-themes not dissimilar to PLwRD—*Challenging/Difficult, Enjoyable, Purposeful*—but sometimes for different reasons. For some, writing was difficult because it reminded them of the better days of their past and also of an uncertain future; but this did not deter them from participating. Writing was also met with enjoyment, reflection, and comfort. How the writing task was responded to varied from “a struggle” to come up with words, taking time to contemplate, to “Basically they were first ones I thought of” (12.7.FTD.BC-1). One informative finding was discovering a two-step progressive response to the prompts. The first response was sometimes characterised as *Challenging or Difficult,* but a second *Reflective* (4th sub-theme) response provided coherence. For others, even contemplating the task of writing was *Challenging*, but after doing so there was a sense of connection not only to the writing task, but to the positive sense it gave about themselves. Writing about aspects of one’s lived experiences with dementia also proved, surprisingly, *Enjoyable*, even if sometimes difficult, across the different prompts. Finding the right words can be challenging but writing can also be stimulating and cathartic, and help to process difficult emotional experiences [59]. A care-partner responding to two poems from prompts one and three (Table 3) connects up her feeling of enjoyment with an appreciation of the economic power of just a few words, as evidenced in Table 5 (see 13.8.FTD.C.1). Writing could equally be *Purposeful.* Instructions for the prompts, asking for one or more words but not longer than a sentence, encouraged a limited but focused response that may have added to a sense of purpose in writing, in part, by reducing the burden to produce many words (e.g., “I felt A sense of relief as silly as that may sound”, 10.2.YOAD.BC-3). This was frustrating for some but also allowed greater accessibility to participate.

#### 3.2.3. Writing Poems: Narrative Synthesis 

Writing about one’s own experiences has been shown to have positive effects on wellbeing [59] but understanding the writing processes that facilitate self-discovery is less known [58]. The process of writing a response to any prompt would likely vary across any group of people, but in developing our approach to research poetry it remained critical to understand how participants responded to the task in order to assess its viability and credibility, evaluate ethical issues, and learn how to improve the process for future practice and research planning. Both PLwRD and care-partners across different diagnostic classifications described positive, enjoyable, beneficial and purposeful aspects to writing. There was a high degree of involvement in the writing process by most stakeholders and this was further emphasised by their responses. Unsurprisingly, for some PLwRD, it was a positive but difficult experience and for others, difficult, challenging, and “hard”, possibly due to changes in cognitive abilities and/or emotions that were triggered. The challenges encountered by care-partners, however, depending on the writing prompt, tended to be around memories of different times and the losses they now feel about changes in the present and anticipate in the future; writing brought those experiences to an immediacy. This, at times, painful reflection, was not rejected and for some welcomed as an opportunity to reflect and engage with emotive thoughts. 

### 3.3. People Living with Rarer Forms of Dementia: Responding to Poems

As discussed above, writing was challenging and sometimes difficult, but reading the completed poems (Table 6) was seen as a distinctive experience that was emotionally connecting, largely positive, and personally meaningful. Each poem is co-authored by a group of people and not a sole individual and the overarching theme, A Collection of Understandings, reflects an assessment of what the poems portray. Two sub-themes, *Has Meaning for Me* and *Staying Connected,* capture the strength of the poems which included being emotive, appreciation for recognition, and feeling less alone, along with acknowledging the poetic process involved in creating. A third sub-theme revolves around *Not Speaking to Me* and focuses on the poetic style of the poem that some participants did not like or understand. How the poem was conveyed also made a difference with implications about delivery format and accessibility “I enjoy the verbal reading of the finished poem far more than the actual written words. It made far more sense in an audio version.” (11.12.FTD.PLwRD-1). 

#### 3.3.1. Care-Partners: Responding to Poems

Responses to the poems (Table 6) across all prompts were mostly enthusiastic and positive in that they Crystallise Lived Experience of dementia and allowed care-givers to experience connection to others, akin to what Stevanovic and Peräkylä describe as experience sharing and emotional reciprocity [60]. A sub-theme, *Bringing Into a Whole*, illuminates the lived experiences of care-partners and struck a particular cord that symbolised the often hidden, unknown and unique worlds of this population. What stands out as a key response was how the poems helped to make care-partners *Not Feel Alone* during a particularly isolating time that included the first and second pandemic lockdowns in Canada, England and Wales. Yet, it was not only the pandemic lockdowns that contributed to feeling isolated; it was also the sense of being alone with a rarer form of dementia where care-partners may not have known anyone else in the same situation. The poems appear to have acted as a catalyst to create personal connections. Although discovering a *Shared Experience* was part of *Not Feeling Alone* for many care-partners, it was also a distinct, powerfully emotional involvement seen as “empowering and encouraging”. A shared experience can be a psychologically powerful phenomena [60], and for some, reading and listening to the poems connected them together with others, decreasing a sense of isolation through shared experiences. Not unlike the two-step progressive response described above in writing, reading poems can also initially trigger painful emotions but, perhaps paradoxically, help people to feel connected through a shared experience. Yet, some felt the poems were *Non-resonating*. This included disliking a poem’s style (e.g., not rhyming or using word repetitions to emphasise and enhance), believing a poem was not powerful enough or did not convey personal experiences adequately.

#### 3.3.2. Responding to the Poems: Narrative Synthesis

It could be argued that the poems were effective because they were meaningful to stakeholders by providing an assemblage of understanding(s) that portray shared and differing emotional responses, provide opportunities for connection to others through shared experience, and help to decrease isolation. The completed poems also highlighted what a group of people can accomplish and create together, even thousands of miles apart. As with most collections of poems, different poems elicit distinctive responses; this is a strength of poetry as an art form [61]. The poems from this study can be seen not as definitive explanations, but as being symbolic of a range of deeply emotional and life-changing experiences. For some stakeholders, certain poems did not personally resonate nor speak to them either because of the form and style of the poem or because it was not emotional enough. For others, for whom the poems did resonate, some had a preference to read the poems, whilst others preferred to listen to the audio recordings. 

We were concerned about overly analysing or deconstructing the experience of responding to the poems; to do so would have subjugated the poems to methodologicism [62] and overshadowed work that is intended to be exploratory, evocative [63], and expressive of what poetry can convey [64]. Yet, understanding how PLwRD and care-partner stakeholders responded to the poems they co-created was important for two main reasons. Firstly, it provided feedback to stakeholders that the poems and their involvement in the research have value [65] and secondly, that the poems have cogency as a form of representation—of their lived experiences—for healthcare professionals, who were involved in the next stage of this project, reported below. 

### 3.4. Stakeholder Group Interviews 

The findings from the thematic analysis of the group interviews were organised into four main themes: Lived Experience and Reflection, Curiosity and Exploration, Barriers and Inclusivity and Sharing the Poems. These themes and their sub-themes (Table 7) are described below and supported with illustrative data excerpts.

#### 3.4.1. Lived Experience and Reflection

This theme captured moments from the group interviews wherein participants noted ways in which the poems helped them reflect on their own experiences. 

*Hearing own experience in a different way* and *Poetry conveys emotion and meaning*

A number of participants, in describing this phenomenon, mentioned the uniquely powerful experience of hearing their own experiences reported back to them in a different way:

“Hearing these words spoken because just inside one’s head is almost not enough [...] It just expands the experience if you hear it. And if you hear it in someone else’s voice [...] because of his inflexions on certain things, and his understanding, and appreciation of those words, it just opened up another voice for me, and I found that—I’m getting goosebumps actually just thinking how that was revelatory to me.” (bvFTD.C-England).


*Validating invisible distress*


Another sub-theme appearing frequently during these conversations was the fact that participants felt, up until this point, that certain elements of their own distress were invisible, and that they were alone with certain experiences. From the conversations it seemed that, in many cases, hearing these experiences reflected back to them in the poems helped them feel validated.

“There was nobody that I could connect with there […] there was no local doctor, even our local Alzheimer’s group, there was no one at all that was even closely in my situation. And when I thought about the idea that other people were pondering this same question at the same time […] I thought, well, I’m not alone. I’m still in a little vacuum. I’m still in my little corner of the world, but somebody else is thinking the same painful thought or […] pondering the same issue that I’m pondering and so I think that was very important for me.” (YOAD. C-Canada).


*Multi-voiced or group lived experience*


Michelle’s account (above) also hints at another theme emerging regularly when participants discussed the relationship between the poems, reflection and lived experience, the fact that the poems, due to the methodology, contained multiple voices, and similar experiences, layered on top of each other. 

“I think it’s fun. I think it’s amusing. It is a lot of fun for me to see how the words were put together. My thoughts and other people’s thoughts. And I see a lot of similarity of course of emotions, flowing throughout the poems.” (PPA.PLwRD-Singapore)


*Poetry conveys emotion and meaning*


Georgia’s account (above), in turn, also demonstrates another theme that emerged frequently when participants discussed the poems in relation to *Reflection and Lived Experience.* This was poetry’s unique capacity to convey emotion and meaning. The words “emotion”, “feeling” and “meaning” came up extremely frequently during the groups, and were frequently, in being brought up, linked together.

“And somehow, the rest of the words put meaning to what you said. [...] It expresses my feelings for me, if you like. I probably couldn’t put it into words. I mean, I can express lots of feelings, but those particular feelings, I couldn’t put- I don’t think, I don’t try, I haven’t tried.” (PCA.C-England)

#### 3.4.2. Curiosity and Exploration

This theme captures responses which foregrounded the idea that research poetry differed from other research activities in promoting a sense of both curiosity and free exploration. 


*Creative aspect piquing interest*


Several responses made reference to the creative nature of the activity, going as far as to suggest this was their main reason for having gotten involved. A care-partner answered a prompt about their reasons for participating: “Just that it was a creative process. I mean, I just love the idea that I could be involved in something. Where I’ve always felt [...] huge support from [blinded], but to actually be able to appreciate something in a creative way was wonderful for me.” (bvFTD.C-England).


*Novel methodology and experience*


There were also a number of similar responses noting a participant’s excitement either at the novelty of the method or the fact that participating represented an opportunity to engage in a new experience:

“Yes, I thought this was a really unusual opportunity and it was really refreshing [...] I thought poetry was a surprising choice [...] it’s the first time I’d been asked to do anything like this and I thought it’s innovative really, and unusual, so I was curious as well.” (PCA.C-England)


*No wrong answers (gut response)*


The final sub-theme of *Curiosity and Exploration* captured contributions which highlighted the ways the poetic form, and its emphasis on there being no wrong answers, allowed individuals to engage their curiosity and explore their “gut responses” in a way other research activities might not have:

“It actually felt good to have participated in something and the process of doing it [...] for me was [...] very creative because it called on emotion, not on cognitive processing and I spend my life in a in different world, and so it was really a stretch to just respond from emotion.” (FTD.C-Canada)

It is worth noting that the above quote, like many of the contributions comprising this sub-theme, illustrated the ways in which the method of narrating experiences without thinking about them too much allowed participants to explore their own emotional life. This theme is therefore heavily connected to the above sub-theme, *Poetry conveys emotion and meaning.*

#### 3.4.3. Barriers and Inclusivity

This theme captured contributions which dealt with research poetry’s unique challenges with regard to engagement. These conversations centred on barriers to engaging with the poems as outputs, but also as a research activity in the first place. 


*Am I a poetry person?*


These contributions tended to put forward the idea that enjoying or creating poetry is an identity trait, with some members of the focus group self-identifying as people who find poetry inherently difficult:

“Just having the opportunity [...] to reflect on how you’re feeling because, my husband and I are both lawyers and we’re very analytical, and so feelings [...] for us or that’s not something that [...] we’re allowed.” (FTD.C-Canada)

Within this contribution, Annie notes the challenge stemming from her and her husband’s analytical predisposition but doesn’t describe it as an insurmountable one. There were also several members, notably, who did self-identify as “poetry people” and for these participants, the poetic aspect made the task easier to engage with than a more scientific research activity.


*Poetry as scary*


Another sub-theme, interconnected with the above one, was centred on the fact that poetry could be inherently scary. This brought together contributions which displayed an awareness, on the part of participants, that poetry has a reputation for being daunting or difficult to understand:

“We have, generally, certainly, in this country, we have a really poor experience of poetry. Perhaps it’s not taught properly at school [...] I don’t know how it materialises, but people run away from it and miss, in my opinion, so much beauty.” (bvFTD.C-England).


*Reading the all the poems was overwhelming*


The final sub-theme within the Barriers and Inclusivity was more centred on the end product, the poems themselves. Many of the contributions under this sub-theme noted that the experience of reading the poems as an artistic output, especially when read all together, felt overwhelming:

“…perhaps highlighting individual poems at different times might be an option. Because it’s quite a lot all together, and that’s one of the reasons why I think it’s been more difficult to share with people, with friends and family, because it’s a whole volume.” (PCA.C-England).

Many of these discussions focussed on the highly emotive material within the poems, and also prompted discussions about other means of dissemination, potentially only asking readers to read one or two poems in one go.

#### 3.4.4. Sharing the Poems

This final theme brought together contributions about what to do with the poems once they were completed and published as a collection. 

*Value as activity* vs. *value as product*

Many of these discussions were prefaced by discussions around whether the end products of the research poetry task had value outside research poetry’s initial value as a research exercise.


*Where and how to disseminate*


Those conversations moved on to wider discussions of how and when to disseminate the poems. Books, dementia websites, radio broadcasts, YouTube and adverts on the tube were just some of the suggestions put forward by participants. Many also suggested that they should be given to medical and health professionals as a means of education:

“I think of educating people [...] it seems like I would be more willing to [...] take the book, say to my husband’s physician office than I would be to give it to one of the people closest to me.” (FTD.C-Canada)


*Risks and benefits of sharing the poems*


One final sub-theme that came up regularly during these discussions was the idea that there were risks inherent to the process of sharing the poems. Many of these discussions focussed on the idea that one might only want to share them with people, such as professionals or really close friends, who were capable of understanding the complex emotions within the poems: 

“Yes, I’ve actually not shared it very widely at all. I shared the book with a friend who is a nurse practitioner actually. I’ve been very careful about who I’ve shared it with, because it’s been an emotional journey really. And I think it’s a lot for people who don’t have experience of any sort of dementia.” (PCA.C-England)

#### 3.4.5. Group Interviews: Narrative Synthesis 

Group interviews, carried out in both the UK and Canada, offered participants opportunities to talk freely about their experiences of a novel methodology, their personal responses to the poems themselves, and possible ways the poems, as outputs, might be shared. The interview findings seemed to indicate that the process of putting the poems together had been a powerful one, providing space to both explore and express complex feelings around individual experiences of dementia care; this was also reflected in the stakeholder surveys, reported earlier. Discussions about what to do with the poems now they had been created almost became a creative activity themselves, with many different (and varied) methods of dissemination tabled and workshopped by the groups. Many participants described their worries ahead of taking part in the research poetry task as well as the rewarding nature of challenging themselves to engage with the activity despite that. The surprising relationship between lived experience, reflection, emotion and meaning making were topics that came up regularly throughout the focus groups. The overall experience of the activity was described by participants as an opportunity to engage in something new, it had given them a new way to look at their experiences, and as a group, to connect to each other through the written word.

### 3.5. Healthcare Professionals and Student Survey 

Throughout the research, poetry process feedback from participants included curiosity as to how the poems would be disseminated including awareness raising or educational opportunities. Our own interest was reinforced and consequently we initiated an additional component of the study to explore the acceptability and usefulness of co-constructed poetry as a learning tool for healthcare practitioners. Although few, similar studies, e.g., [66,67], have suggested that poetry, as a supplementary learning tool, offers students an opportunity to reflect on more holistic views of lived experience, furthers their understandings and ability to engage in critical thinking, and stimulates changes in learner disposition (i.e., emotion). Thus, poetry may be valuable to address both Bloom’s taxonomy of cognitive learning [68] and Krathwohl’s taxonomy addressing the affective domain of learning [69]. The cognitive learning taxonomy, hierarchal in presentation, identifies four types of knowledge and related skills that learners are expected to acquire: factual (i.e., describe learning), conceptual (i.e., explain learning), procedural (i.e., knowledge application) and metacognitive (i.e., analyse, evaluate or prioritise knowledge) [70]. However, in affective learning, the focus is on the development of characteristics that shape thinking and behaviour. These might include, for example, compassion, advocacy, or respect [71]. As Schmidt [72] and others argue, emotions are a critical ‘force’ for any meaningful knowledge acquisition. Also hierarchal in presentation, affective learning or skills includes receiving or basic awareness, responding to the learning, evidence of valuing the information received and organising value systems, and finally demonstrating values as impacting behaviour [69]. 

The sample is set out in Table 2. After cleaning the data (*n* = 101), we obtained a final sample of 93 participants (professionals, *n* = 55, and students, *n* = 38) who completed the survey. The majority of participants were female (n = 69), not uncommon in the professions represented, and between the ages of 30–49. Students were significantly younger, however. Interestingly, most of the participants had experience (practice and/or personal) with people living with dementia (*n* = 62), and only 27 reported either reading or writing poetry within the last 5 years. Table 8 provides survey group composition, sample size, and selected poems (see Appendix A for the poems). Our thematic analysis was both deductive and inductive. We first created a codebook informed by our interest in both cognitive and affective learning and critical thinking, and then added emergent codes during the coding process. This was followed by the researchers engaging in analytical discussions to identify relevant themes across the data [36,37]. 

It is worth noting that responses to the open-ended questions ranged from one word to over 300 words. Whilst word quantity does not confirm engagement or quality learning, arguably the poetry appeared to offer an external voice for people living with dementia and provoked or “opened a space” [73] for reflection on either practice or, in some instances, personal experience. Our thematic analysis of the survey’s open questions captured three themes specific to learning: (1) other-oriented perspective taking; (2) insight through critical self-reflection; and (3) value of dementia support. These themes corresponded with the lower to mid-order learning levels on the cognitive and affective learning taxonomies, although we recognize that these are not necessarily discrete categories [69]. We also acknowledge that these are learners’ brief written expressions of understanding rather than observed changes in behaviour because of their exposure to new knowledge [74]. 

#### 3.5.1. Other-Oriented Perspective Taking

It was strongly evident that the poems provided participants with an opportunity to empathetically connect with the varied descriptions of lived experience set out within the poems. Participants in each group expressed either “imagining” the impact of experiences on the person affected by dementia and/or “feeling” similar emotions to those expressed. Whilst each poem conveyed different content on the dementia experience with some being more negative in tone than others, the poems did not vary greatly in relation to generating empathetic responses. However, some poems seemed to vary in terms of generating basic awareness of emotional content (lower-level learning) such as, “I felt the fear of the speaker” (student, Group 1), versus those where participants appeared to value the message inherent in the words (mid-level learning). For example, a healthcare professional wrote an ability to feel a “mixture of emotions” in addition to valuing the contrasting emotions and meaning in the concluding message in reference to the poem There is So Much I Could Say:

“In this poem I felt a mixture of emotions. At the start, I felt so deeply sad about the impact dementia can have on every part of life. In response to the line, ‘a slow and painful death I encounter each day’, I thought about how cruel dementia is as a disease, in which people feel themselves slowly dying and how relentless this may be. Towards the end of the poem, I felt lighter in myself whilst reading the verse ‘My husband is my best friend and soulmate. I am a lucky woman to have him in my life. He treated me like a queen for over 30 years and he is still my knight in shining armour.’ It highlighted the person, instead of just the diagnosis and I smiled at the love that was so evident in this part of the poem. I was left really thinking about the instruction to ‘Live each day as that is the only guarantee you have’ and to take life as it comes, not worry about the trivial and enjoy as much as I can.” 

#### 3.5.2. Insight through Critical Reflection

Closely associated with the poems’ generation of empathy were participant responses that demonstrated critical reflection on new learning and/or affirming existing knowledge. Overall, our findings here are consistent with Speare and Henshall [73] who, using poetry, also identified the value of reflection in relation to a practitioner’s identity and the possibilities within their role. Responses here appeared to provide higher-order learning including the participant understanding, sometimes analysing the knowledge, and active application of the knowledge in future practice. For professional participants with considerable work experience comments were sometimes more fulsome or detailed. Student participants were more apt to reference the personal experience of a grandparent living with dementia.

In terms of reflections for direct practice, a healthcare professional drew attention to active communication skills and person-centred approaches from There is So Much I Could Say:

“I think this poem reminded me to be patient when working with care recipients. There are numerous reasons why a person cannot articulate themselves well. It may or may not have to do with the dementia, but there is power in listening. Allowing the client the floor. Shows a person-centred approach to care and that your time is not more important than their experience.” 

A healthcare student expressed their learning from Still Me, with a focus on dignified approaches to care and support:

“Always being mindful of sensory overload, personalized care focusing on the patient’s strengths and abilities. They are trying so hard not to be angry with themselves we MUST try too. Unfortunately, I have seen too many ‘professional’ caregivers become angry or frustrated with their patient’s/client’s limitations, this is unsupportive, uncalled for and disheartening. An example of what never to do.” 

Importantly, for some participants there was critical engagement in terms of reflections on the limitations within the current arrangements for dementia care and support for people who had atypical types of dementia. A student stated the following in response to reading the poem Tumble Jumble:

“Overall, the poem reminds me of a concept called ‘medical waiting’ that I learned of in an article by Mary Hunter called ‘The Waiting Time of Prostitution: Gynaecology and Temporality in Henri de Toulouse-Lautrec’s Rue des Moulins, 1894. Medical waiting is about the anxiety one experiences while waiting for results of medical testing, and the fears surrounding the way the body will be policed or regulated by institutions after the diagnosis is delivered. To me, it seems that the impact of institutions—with their treatment protocols and the way they regulate, limit, and treat patients (both medically and interpersonally)—on the bodies, minds, and overall wellness of those being diagnosed…”

#### 3.5.3. Value of Support

Tailored learning with respect to people living with a rarer form of dementia was evident, however, understandings specific to support and its value for anyone affected by a rarer dementia diagnosis were much more prominent. For example, When Will I No Longer Be Me stimulated the following comments from one professional which seemed to convey their responsibility to do something about support such as a village of love, acceptance or kindness: “empowerment to create understanding around dementia let’s build that old-fashioned village again. Surely it must be our role to create this old-fashioned village of care and community for people with YOD [young onset dementia]”. Similarly, characteristics of support were emphasised by a student in response to Accept this Honour: 

“As with the other poems, when caring and support dementia patients, compassion and patience is key. They are not themselves, their behaviours if they have any are not who they are but a product of their disease progression and deserve compassion, love and patience.”

The poem Support generated many concrete examples of what support is necessary and captured by a student who wrote about support for the family as opposed to the more traditional focus on an individual system: “This poem describes the importance of supporting both the person with the diagnosis but also their family members as it is important to see everyone in that circle of care as important and needing support through a difficult diagnosis.” 

#### 3.5.4. Acceptability of Poetry as a Learning Tool

Although most participants did not read or write poetry, its use as a learning tool was both favourably received and encouraged—and in some instances, responses included recommendations on how to improve their use. There were no responses that discouraged the use of poetry for learning. Poetry appeared to be more accessible than traditional academic reading:

“Articles from academic journals can be so stale—clinically reporting the experiences of others and it has us looking down at them and their struggles. However, qualitative/artistic/narrative can actually have us be with the person, their experience and their struggles—have us empathise to the point of us being in their own shoes, and better understand how we can interact with them as professionals in a way that’s more humane.” (healthcare professional)

The enhancement of empathy through poetry resonates with Muszkat et al. [75] who used poetry with medical students to understand the lived experience of being a patient.

Building on poetry’s accessibility or ability to “be in another’s shoes”, participants also offered us suggestions to extend the learning reach:

“…if it is integrated into the curriculum/supervision in ways that highlight specific issues and experiences (e.g., isolation, fear of the future, not knowing what’s next, dementia activism, different symptoms and problems). I wouldn’t just bring in poems and read them without a context though. It would be important to use them (as visual art is now used in some medical school training) to help support whatever teaching or supervision is occurring---otherwise eyes will roll and only a few interested trainees will take it on. Have the class create poems together, that’s where I might begin.” (healthcare professional)

#### 3.5.5. Using Poetry as a Learning Tool

Our additional investigation to explore how poetry could be used as an approach to learning with health care professionals offered additional insights for research poetry. The poems appeared to engage participants as learners for gaining insights on both the lived experience of rarer dementias as well as implications for practice. Our survey offered promising findings with respect to cognitive and affective learning around empathy, critical thinking and the conceptualization of what constituted support. Whilst the survey had limitations in terms of it not being an interactive learning tool or used in conjunction with other learning methods, it was well-received by participants and its use encouraged for future work on educational approaches for dementia practice.

## 4. Discussion

### 4.1. Cross Synthesis and Interpretation 

This is the first study we are aware of that examines how poetry, co-constructed by PLwRD and care-partners, can inform healthcare professionals’ understanding of living with rarer forms of dementia, and how it might contribute to future training and education. The study also provided an in-depth examination of the process of co-constructing poems and assessing the impact of research-inspired poems on stakeholder–participants before soliciting responses from healthcare professionals. We learned that poetry can be meaningfully co-constructed electronically at a geographical distance by PLwRD and care-partners, offering new public health options to engage people regionally, nationally and internationally in research and in peer support groups. The effect of the research on stakeholders, which is rarely reported, was qualitatively assessed, and we established the validity, trustworthiness and value of the poetry-creating process and its resulting output; this was essential to do prior to surveying healthcare professionals. 

The methodological approach to creating poetry used in the present study sought to “create a voice beyond (one) individual” [11] (p. 36), thus inviting multiple voices of those impacted by dementia —care-partners, bereaved carers and PLwRD—to shape the research. This seemed particularly cogent considering the isolation experienced by this population and the lack of information and knowledge available about rarer forms of dementia among healthcare professionals and the general public. Likewise, we also explored if this form of creating poetry could be a viable arts-based qualitative methodology [76] by using participant-voiced co-constructed poems as experimental texts to “evoke the reader’s emotional response and produce a shared experience” [77] (p. 1330) within a research context. After poems had been created in the study’s first stage, the second stage used stakeholder surveys to discover what PLwRD and carer-partners thought about the creative process of making the poems and what they thought about the completed poems. This, together with researchers’ thematic interpretations of the words/phrases making up the poems, created a balance between participant meaning and researcher interpretation, “a balance that relies heavily on both subjectivity and reflexivity” [78] (p. 580); this balance was central to our methodological approach. 

Going through uncharted territory where there is a lack of clinical and support options differs from the experience of older people who most often experience memory loss as the first and primary dementia symptom after the age of 65, e.g., [79]. The poems and surveys highlight that attempting to access often non-existent support from healthcare, charities and some families was a continual challenge as symptoms continue to progress and personal needs continue to evolve. Through their poems, stakeholders informed us that some professionals were slow to respond, unknowledgeable at times, and misunderstood symptoms that led to misdiagnosis, and a further sense of going it alone. This was mitigated by those who encountered professionals who were knowledgeable and provided appropriate care and/or referrals, supporting work by Stamou et al. [80] and Woolley et al. [81] who, respectively, call for the development of specialist services and awareness of mis-diagnosis. 

Contributing to poems through the sharing of their lived experiences, support needs and opportunities for support provision, were seen as both relational and an action. Theoretically, the poems convey that support was conceptualised as relational [82], whether with family, friends or professionals [44], through the sharing of knowledge about their type of dementia and the desire for compassionate exchanges that attended to their feelings and emotions, including both hopefulness and distress. Tranvåg et al. [46] describes this as a form of dignity, “strengthening their ‘relational self’ as a source toward enhancing dignity experience” (p. 589).

Interestingly, for stakeholders, there was no distinction made between support received from family/friends or professionals, highlighting the vital importance of social and emotional peer support networks [83,84,85], alongside that which is provided by the formal support sector. Support as an action appeared to involve indirect or bi-lateral exchanges through deliberate gestures of help for others as well as something received from others. Support seen as an action also appeared to challenge the notion of passive recipients of assistance [79]. Even in the most ominous personal circumstances where support services were typically not available, adaptive mechanisms and/or active coping appeared to be present, highlighting the resourcefulness and resilience of those navigating these uncharted territories. Seen also as a “form of co-cooperative communication, co-cooperative action and co-operative care” [86] (p. 1167), there was a desire by participants to inform and influence how support is organised and delivered. 

### 4.2. Limitations and Strengths

Not having access to a computer or other electronic devices and dependence on adequate Internet connectivity may have limited stakeholder participation. A lack of interest, intimidation about poetry, or feelings of uncertainty about what poetry might offer could also have deterred participation. Although only voiced by two stakeholders, other possible deterrents may have been related to concerns that this activity might bring up disturbing memories and emotions that are not resolved or quieted by the poetic process. The language of the study was limited to English or Welsh and most participants were from a Western culture. The stakeholder portion of the study took part during the first two years of the COVID-19 pandemic and it is possible that some responses were shaped by this health crisis. The group interviews and stakeholder surveys also illuminated possible barriers to engaging with research poetry. These include earlier educational experiences of poetry being badly taught [87], assumptions that poems have hidden meanings [88], being unsure about what a “non-poet” could contribute [89], and more traditional understandings of how poetry should be structured. To address these possible barriers, in future research we suggest that more detailed information be provided at the onset of a project about the possible range of poetic styles and differing structures (e.g., there is no one typical or traditional poetic form and poems do not need to rhyme), further information be provided about the facilitating poet’s creative process, and to more strongly emphasise that previous poetry experience was not required; these points may need to be reinforced throughout a study that draws on multiple contributions over time. Regarding healthcare professionals’ involvement, we were not able to recruit a larger sample size and current participants should not be seen as representative of their respective professions. Although designed to minimise demands placed on professionals, an online survey may not have been the most advantageous recruitment approach. 

Creating research poetry electronically went surprisingly well and feedback solicited from participants after the completion of each poem confirmed it was a feasible approach to engage this population. In addition, it was seen as a positive and helpful experience even though it focused on very difficult life-changing conditions. Survey responses also saw the poems as offering valuable information for healthcare practitioners unaware of rarer forms of dementia, and possibly the general public. The time taken for participating was also far less than in interview-based research, making this approach to research poetry an accessible option to use across different stages and types of dementia with different cognitive and physical problems. Likewise, unlike in most interviews, stakeholder–participants were given more time to respond (up to two weeks) to prompts and the option to use only a few words to respond that need not be grammatically correct or in complete sentences. In addition, the avoidance of a back-and-forth question-and-answer environment for the creation of poems and survey responses, that can prove to be cognitively challenging and emotionally stressful, was seen as a positive and novel contribution. The use of small cohorts (4–13 people) over about 12 weeks with three distinct prompts, allowed different information to be economically solicited with minimal inconvenience reported by participants. 

We carefully considered whether to include both PLwRD and care-partners in developing the poems. Although those diagnosed with dementia will likely have a different embodied experience than a family member or close friend who provides support, there is most definitely a linked and a shared experience “with each party being integral to the other’s experience” [90] (p. 709). Both PLwRD and care-partners—from the time of first noticing “something was wrong”, to the challenges of obtaining diagnosis, changes in employment and finances, and experiences of the lack of healthcare and support options—reported being profoundly affected by these lived experiences. We felt it experientially and theoretically important to keep their voices together as a shared narrative [33,34,82] in the completed poems, while identifying contributions through direct quotes in the results section above. We also wanted to involve the contributions of both PLwRD and care-partners in order to ensure a balanced representation of experience across dementias, where otherwise cognitive challenges experienced by specific diagnostic groups might have limited their viewpoints being captured. This decision was also influenced by four of our authors, who had been dementia caregivers. They felt that the words of PLwRD and care-partners be used together within the poems to better reflect, poetically, lived experience. Involving both the PLwRD and care-partner contributions to the poems also reflects how dementia services often provide support and care for those impacted by dementia. For stakeholder surveys, however, PLwRD and care-partners’ responses to writing and responding to poems seemed better suited to be analysed separately in order to more fully understand their respective participatory research and creative experiences [15]. As a scientist who regularly uses and teaches research poetry, Illingworth [61] is clear that a poem does not have “any single true meaning…it is not an objective certainty, its interpretation is largely subjective” (p. 52). “When you read a poem, you do so while simultaneously bringing all of your own lived experiences, knowledge, and identity to your interpretation, or analysis, of the poem” [61] (p. 17), and this is a methodological strength, we would argue, of this study.

### 4.3. Future Research and Practice Implications

This research has demonstrated that poetry can effectively be co-constructed electronically by people living in diverse geographical settings over time. This opens up support opportunities for people who live at a physical distance or who are unable to leave their homes to participate in meaningful creative activities. Health authorities, dementia charities, regional governments, experienced poets, and others who offer dementia support can feel confident that electronic forms of poetry can be organised and delivered efficiently to people who may be overlooked or unable to otherwise participate. Although this study was conducted entirely electronically, in-person support groups could also be involved in sharing words to create poetry, which could be used reflectively and to document a group’s evolution over time. The contributions of the poets involved in this study were invaluable. Their knowledge of poetic forms and their interest in learning about different forms of rarer dementia led them to place the voices of participants as their primary focus in creating the poems. These are important considerations in developing co-constructed poetry practices involving poets and people impacted by dementia. 

This is the first research study that we are aware of that has involved poetry as a means of portraying PLwRD, care-partners and bereaved carers’ lived experiences for healthcare professionals. This has implications for the training and education of these professionals that extend past traditional classroom and clinical learning practices. Future research should consider how poetry could be used to engage groups of healthcare professionals and students to communicate about caring for people with rarer forms of dementia [91]. Likewise, devising research that uses poems composed by groups of people with different types of dementia, e.g., [92]—as a way to understand dementia more broadly—could provide an adjunct or alternative to the ‘traditional’ qualitative research interview. From a public health and policy perspective, further understanding the health, wellbeing and social benefits of sharing personal experiences, through reading and listening to research-generated poetry, would also be an opportunity to expand public engagement research in dementia and offer new co-created research opportunities [15,93]. 

## 5. Conclusions

Whilst poetry and research may not appear to fit ‘naturally’ or comfortably with each other, this study has shown that poetry co-constructed by those impacted by rarer dementia together and experienced poets, can be effectively created electronically and at geographical distance. The study has also shown that this form of research poetry can contribute to healthcare professionals’ understanding of rarer dementia, whilst also providing creative and reflective engagement in research. And equally important, it was seen by most participants to be a positive, helpful, interesting and accessible approach that had value and meaning for them. Welcoming participants to be co-creators and evaluators to develop poems, as has occurred in this study, can be an empowering experience [21]. As an intervention and methodology, poetry can be examined within a multidisciplinary context to provide multiple opportunities for research, provide a tangible and meaningful outcome for research participants in the form of co-created poems, and thinking ahead, from a public health perspective, offer potential opportunities to engage the public with hidden and lesser-known health problems. 

## Figures and Tables

**Figure 1 healthcare-12-00485-f001:**
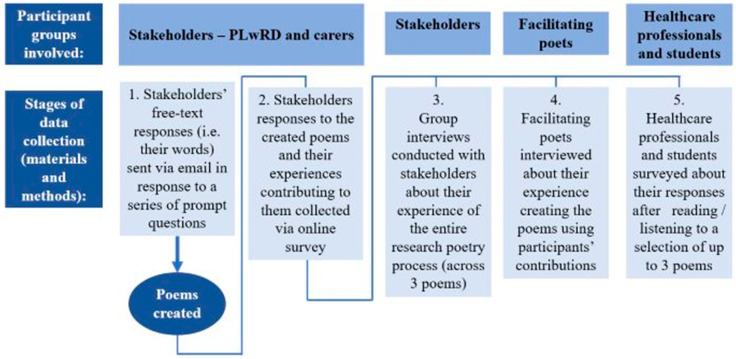
Data collection.

**Figure 2 healthcare-12-00485-f002:**
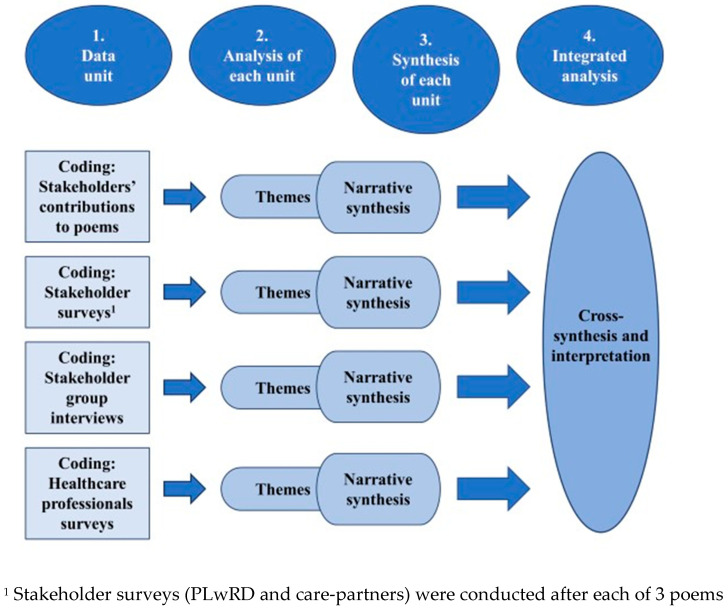
Data analysis process.

**Table 1 healthcare-12-00485-t001:** Stakeholder demographic information (*n* = 71).

Variable	*n* (%)
Age ^a^ (Mean = 63.0, SD = 9.8)	
Gender ^b^	
Female	47 (70.1)
Male	20 (29.9)
Diagnostic subtype *	
PCA ^1^	26 (36.6)
FTD ^2^	22 (31.0)
PPA ^3^	9 (12.7)
YOAD ^4^	7 (9.9)
Mixed dementia	4 (5.6)
Other dementia ^5^	3 (4.2)
Status	
PLwRD ^6^	27 (38.0)
Care-partner	39 (54.9)
Bereaved carer	5 (7.0)
Relationship to PLwRD	
Spouse	33 (75.0)
Child	8 (18.2)
Other relative	3 (6.8)
Ethnicity ^c^	
Caucasian	63 (92.6)
Asian	4 (5.9)
Métis ^d^	1 (1.5)

^a^ age not identified = 4, ^b^ gender not identified = 4, ^c^ ethnicity not identified = 3, ^d^
https://indigenousfoundations.arts.ubc.ca/metis/(accessed on 15 October 2023); ^1^ posterior cortical atrophy, ^2^ frontotemporal dementia (including familiar FTD (fFTD) and behavioural variant FTD (bvFTD), ^3^ primary progressive aphasia, ^4^ young-onset Alzheimer’s disease, ^5^ Lewy body dementia (LBD) and young onset vascular dementia (YOVaD), ^6^ person living with rare dementia, * inclusive of care-partners and PLwRD per type.

**Table 2 healthcare-12-00485-t002:** Healthcare professionals and students in training demographic information (*n* = 93).

	Professionals	Students
**Sex**		
Male	13	7
Female	41	28
Non-Binary	0	2
Prefer not to say	1	1
**Age**		
18–29	4	31
30–39	15	6
40–49	15	0
50–59	11	0
60+	10	0
**Location**		
Canada	1	38
United Kingdom	50	0
United States	3	0
Other	1	0
**Occupation**		
Researcher	5	0
Psychologist (Clinical)	18	0
Nurse	7	0
Psychotherapist/Counsellor	2	0
Occupational Therapist	4	0
Social Worker	2	0
Neuropsychologist	3	0
Paid Caregiver	1	0
Other	8	0
Honours Bachelor of Social Work Student	0	18
Bachelor of Science Nursing Student	0	20
**Dementia Care Experience**		
Yes	44	18
No	11	20
**Types of Dementia Experience**		
Frontotemporal dementia	22	13
Young onset Alzheimer’s dementia	33	12
Lewy body dementia	31	15
Posterior cortical atrophy	15	0
Familiar Alzheimer’s disease	24	13
Familiar frontotemporal dementia	14	6
Other	12	4
**Read or Wrote Poetry in last 5 years**		
Yes	14	13
No	41	25

**Table 3 healthcare-12-00485-t003:** Themes across poems organised by prompt.

Prompts	Themes/*Subthemes*	Associated Poems ^1^
Prompt 1: What is your experience of having (type of dementia) or being a family member caring for someone with this type of dementia?	Temporality Disrupted *Uncharted Territory* *Being in the Moment* *Rediscovering Identities and Relationships*	(1) A Steep Learning Curve (2) Still Me (3) No One Would Believe Me (4) Changing (5) Eleven (6) Accept This Honour (7) When Will I No Longer Be Me? (8) Out Here in The Rain (9) How May I Help You?
Prompt 2: How would you describe (type of dementia) to a friend or family member?	A Rollercoaster *Heart-Breaking Losses* *Shifting Selves and Changing WorldsMisdiagnosed and Misunderstood*	(10) Bewilderingly So (11) Where to Begin (12) Unwavering Braveness (13) There is So Much I Could Say (14) Description (15) I Know You Have Noticed (16) Tumbling, Jumbling (17) Why I Repeat the Same Question at Random (18) Who Our Loved Ones Used To Be
Prompt 3: What does support mean to you?	Multidimensional Support *Not in This alone* *Compassion and UnderstandingSupport as an Action*	(19) Truth: A Fib (20) SUPPORT (21) What Else Could I Try To Do (22) A Journey (23) Relief (24) Help Me (25) To Continue (26) The Resiliency (27) Please Stick Around

^1^ Appendix A contains all poems.

**Table 4 healthcare-12-00485-t004:** Themes, sub-themes and illustrative quotes.

Themes/ Prompts	Sub-Themes	Illustrative Quotes
**Temporality** **Disrupted**/What is your experience of having (type of dementia) or being a family member caring for someone with this type of dementia?	*Unchartered Territory*	“A journey into the unknown, caring in love, the constant challenge of adapting” (6.5.PCA.C-1); “Doors closing, fading slowly. Becoming invisible and written off” (9.12.FTD.PLwRD-1); “uncharted territory, cascading loss, fear and uncertainty about what’s coming next” (13.5.PCA.C-1); “The sister I loved so dearly was being taken from me, piece by piece” (10.2.YOAD.BC-1); “Our dreams are shattered and we will never be the same again, I never thought a disease could be so horrible and have such a dramatic impact on a family” (13.11.FTD.C-1); “Losses of myself, my essence, my work, my driver’s licence, my job, my body, friends, love of husband” (13.12.FTD.PLwRD-1); “I’m stuck here with a rare *disea eeeze* and the physicians don’t have a clue” (12.5.LBD.C-1); “professionals [who] seem at a loss and are slow to respond” (13.9.PCA.PLwRD-1)
	*Being in the Moment*	“Time slows down and remembering to truly be in the moment” (5.1.PCA.BC-1) “Trying to be positive and know these moments with him won’t come again” (10.4.MD.C-1) “Take every opportunity to have a laugh” (6.17.PCA.PLwRD-1)“At first it was a hectic pace, as time passed I felt more at peace that I am doing the best I can do” (8.3.FTD.C-1).
	*Re-discovering Identities and Relationships*	“Then curiosity and call to action took over. I have been learning about myself and a brighter side of this world ever since. I am energised and happy” (11.10.PPA.PLwRD-1); “I needed to accept what was what and become a better person. I needed to fight, be more patient and find answers. (8.3.FTD.C)
**A Rollercoaster**/How would you describe (type of dementia) to a friend or family member?	*Heartbreaking Losses*	“heart breaking” (10.3.YOAD.C-2; 12.7.FTD.BC-2; 9.10.FTD.C-2); “frightening” (9.16. FTD.C-2; 6.23.PCA.PLwRD-2); “The Destroyer of Lives” (9.11.FTD.C-2); “the cruellest dementia you have never heard of” (9.3.FTD.C-2); a “rollercoaster” (12.7.FTD.BC-2; 5.8.PCA.BC-2); “constant emotional and physical ups-and-downs” (8.3.PPA.C-1) “She is still the same person, but is not able to do the things she used to do because of the disease” (13.10.PCA.C-2); “Not able to make any decisions big or small on most basic tasks. Choking often, falling, loss of balance. Loss of not being able to accomplish what used to be second nature” (13.12.FTD. PLwRD-2); “Brain blindness. Extreme turmoil. Unwavering braveness. An excruciatingly slow drip of profound loss” (7.7.PCA-C-2).
	*Shifting Selves and* *Changing Worlds*	“A disease that…erodes everything you thought about that person and your relationship with them. A disease that eats away the soul of the person and everything they are or who they had wanted to be.” (9.10.FTD.C-2). Profound changes in how the self relates to the world created a sense invisibility, “Increasing opacity between me and the world” (11.5.PPA.PLwRD-2), “You just don’t see me anymore” (12.4.YOAD, PLwRD-2); “retreating into own world” (9.4.FTD.C-2).
	*Misdiagnosed and* *Misunderstood*	“Things aren’t where they appear to be. Which way round does this go? Where is the top or the bottom?” (6.5.PCA.C-2)“A neurological disease that does not follow the perceived norms of dementia of memory loss and only for the older person but far more aggressive and devastating as it affects behaviours and personalities of younger people in ways that make them disinhibited, lacking empathy and insight” (9.10.FTD.C-2); “a misdiagnosed and misunderstood condition” (6.22. PCA.C-2); “they never say drop him off to my house and get your hair done … they just say LOOK AFTER YOURSELF” (9.20.FTD.C-1)
**Multidimensional Support**/What does support mean to you?	*Not in This Alone*	“Support means not feeling utterly alone. It means having someone to help bear the unbearable” (P7.7.PCA.C-3); “Meeting others who are in the same place as you and who understand” (P9.10. FTD.C-3); “a togetherness beyond family” (P5.1.PCA.BC-3); “Support helps me when I get confused and frustrated…it helps me live my life independently” (P6.1.PCA.PLwRD-3); “Being part of a PPA network, not feeling alone…giving me a degree of comfort” (P11.1.PPA.PLwRD-3); “Subject matter experts who listen objectively and share their expertise” (P8.3.FTD.C-3); “Professional help to educate doctors and the general public” (P13.12.FTD.PLwRD-3); “Provide support strategies that are specific to my condition” (P13.9.PCA.PLwRD-3) “Others understand or have travelled this road as well” (P13.2.YOAD.C-3).
	*Compassion and* *Understanding*	“Support means you are aboard my train and will help in any way you can” (P8.4.FTD.C-3); “Taking time to listen and giving time for talking (P13.PCA.PLwRD-3); “In between there is support that helps me see things differently and support that just takes time to listen and see things the way I do” (P10.3YOAD.C-3); “Old fashioned underwear” (P6.8.PCA.C-3); “From someone to hug me to someone who can take over when it’s too much. Someone who listens….and believes me” (P10.1.FTD.C-3); “Learn to accept me as I am, with all my faults and errors” (P12.4.YOAD.PLwRD-3); “Not judged by expression of my thoughts or feelings” (P10.2.YOAD.C-3); “Support me by showing acceptance” (P12.6.VD.PLwRD-3); Help to affirm decisions” (P5.2.PCA.C-3).
	*Support As An Action*	“Always include stimulation. Always include companionship” (P12.3.FTD.PLwRD-3); “Support means enablement/kicking ass/being there when down” (P6.14.PCA.PLwRD-3); “It lifts me up in times of doubt. It allows me to move forward in a positive way with my life.” (P11.12.FTD.PLwRD-3); “Unconditional actions to ease the burden of care” (P12.7.FTD.BC-3); “If I am able to connect/talk with others that have PPA and I am able to help/support them—that is very helpful for me” (P7.4.PPA.PLwRD-3); “Support is helping me to feel better about myself as I contribute to help others.” (P7.4.PCA.C-3)

**Table 5 healthcare-12-00485-t005:** People living with rarer forms of dementia and care-partners’ experience of writing by theme with illustrative quotes.

Stakeholder Group	Themes	*Sub-Themes*	Illustrative Quotes
**PLwRD**	I Wanted to Do It	*Challenging but* *Enjoyable*	“I am much more comfortable with the written word than I am with verbal conversation and that carries over to writing poetry”; (12.4.YOAD.PLwRD-1), “I found it very easy to just sit quietly and let my thoughts form my words” (12.6.YOVaD/.PLwRD-2), “I found finding the words was relatively easy as it was around the basic support that I feel I need on an ongoing basis (11.12.FTD.PLwRD-3)”; “Challenging but nonetheless enjoyable” (11.1.PPA.PLwRD-2); “I found the first word was difficult and challenging, but found after the first word was down, the rest just seemed to come easily, and it was then enjoyable” (6.1.PCA.PLwRD-2); “Having never written poetry before I found it challenging but a positive thing to do” (6.7.PCA.PLwRD-3).
		*Hurts So Much*	“I had a very hard time understanding the question and accessing the words. I felt very detached” (13.3.YOAD.PLwRD-2); “Sensed that I was having problems, Frustrated, when can’t remember” (8.1.PPA.PLwRD-2); “Challenging and to start and scary to go in your feelings cause I try to suppress how I really feel cause I’m afraid to go negative and think of all that hurts so very much but it ends up feeling so good to get it off your chest and ends up being cathartic. So much that I would like to go deeper and write more and get out my sad feelings instead of pushing them so deep and these 2 exercises have made me realize how good it is to see it on paper and hear it read by someone. It’s such a good idea.” (13.12.FTD.PLwRD-2).
		*Purposeful*	See Section 3.2.1 in the text
**Care-Partners**	Consolidating	*Challenging/Difficult*	“Challenging to distil my thoughts in just a few words” (6.5.PCA.C-1) “I struggled a bit to decide on what words really captured my experience” (10.3.YOAD.C-3, “First was the battle of just getting to the task but to actually sit down and consolidate the thoughts into sentences was met with great reluctance” (13.10.PCA.C-2).
		*Enjoyable*	“It was easy and enjoyable for me. The words popped into my head quite readily. It was interesting to me to realize how much can be said with just a few words (13.8.FTD.C-1); “The experience was enjoyable because I think of many words that mean ‘support’. It was cathartic to put those words on paper, and I felt less isolated and more supported knowing that other people were doing the same thing” (13.8.FTD.C-3).
		*Purposeful*	“It was not a challenging task because I immediately knew what I wanted to say and I found it exciting to add to the gut-level offering I knew it would become…My staccato words express the small bites at life which is our daily experience; one thing at a time is the only way to survive this.....for us.” (9.11.FTD.C-1); “…forced me to truly think about what I would most want a friend or family member to know about the diagnosis from my perspective as care partner to my wife. I believe I captured that well” (8.4.FTD.C-2).
		*Reflective*	“In the end I decided to share some of the common themes I would tell others during the experience. It was touching to remember those days and to reflect on how I feel now versus how much harder it was when she was alive” (10.3.YOAD.BC-1); “It was helpful in causing me to actually sit down and reflect on what we wanted (needed?) people to know and how to interact” (13.10.PCA.C-2).

**Table 6 healthcare-12-00485-t006:** People living with rarer forms of dementia and care-partners responding to completed poems by theme with illustrative quotes.

Stakeholder Group	Themes	Sub-Themes	Illustrative Quotes
**PLwRD**	A Collection of Understandings	*Has Meaning for Me*	“I love the completed poem, it has a meaning to me as I can relate to the words written down, and it’s interesting to see how a poem can be thought of by a list off words or sentences by different people” (6.1.PCA.PLwRD-1); I liked the way our various views and feelings were interwoven with other people’s responses to give an active description of the challenges of life living alongside dementia” (11.12.FTD.PLwRD-1); “I appreciated being able to describe some of my feelings about being ill and the sadness about no treatment/cure” (7.4.PPA.PLwRD-1); “It touches a chord for me. I absolutely like it and feel less alone in this…now I rarely write but when I read something good I’m amazed at the craft and how people put the beautiful words together and convey images or feelings succinctly. I can’t get it out right and it hurts that I’ve lost it but this makes me want to try again. Thank you! Great idea” (13.12.FTD.PLwRD-2).
		*Staying Connected*	“The poem helps to capture the different ways people are thinking about what support should be for them. The words or the ways of saying may be different, but the emotions and concerns behind those words are very similar. Clearly articulates the emotional and personal aspects of what are perceived as needs—caring, communicating, staying connected, kindness—seem to override the many physical supports required although they are just as important for the overall well-being of the person and their care giver)s)” (13.9.PCA.PLwRD-3); “I liked the idea that the words were formed into a virtual buttress to allow us to fight back and defend us from the trials and tribulations that our condition now causes us in our lives.” (11.12.FTD.PLwRD-3).
		*Not Speaking to Me*	“Sorry, did not sound like a poem—but maybe I am not up to date on how poems are now/can be. My words were taken fine. Very short comments—thought they could be elaborated more” (7.4.PPA.PLwRD-1); “It was more like a statement than a poem. It read a little bit like a poem, but much more like a book” (11.1.PPA.PLwRD-1); “To me with my understanding of what a poem is I did not find it to be a poem just a collection of words” (12.2.YOVaD.PLwRD-1); “I did not appreciate the poem. It seemed a little cumbersome and though I understand that these words were contributed by group members, I do not feel that the author expressed our deeper feelings” (11.10.PPA.PLwRD-2).
**Care-** **partners**	Crystallise Lived Experience	*Bringing Into a Whole*	“I really liked how it started out in the first line. FTD is indeed a journey. Then....beginning each stanza with the words “There is so much I could say...” Then, you HEAR this spoken from the perspective of each care partner. Each stanza stands on its own and is unique, which, to me, speaks to the disease itself. I very much treasure the final product and the creativity to bring individual parts into a whole” (8.4.FTD.C-2); “Beforehand was cynical about how a poem could be constructed from a collection of words submitted by various contributors. The final poem crystallises the various feelings, emotions and experiences of rare dementia carers” (5.8.PCA.BC-1);“Both my husband and myself really loved the finished poem and felt it expressed so clearly what it feels like to suffer from PCA. The title is excellent and the repetition of the first and last lines in each verse is particularly effective” (6.5.PCA.C-2); “Spoke volumes to me! It is amazing how different people’s thoughts could come together to create such a symbolic poem of dementia” (10.2.YOAD.BC-1).
		*Not feel alone*	“Reading the pain and hope of others makes me feel connected, and this disease typically causes such rifts of disconnection” (10.5.PPA.C-1). Initially I felt overwhelming sadness and grief, while at the same time it’s helpful to know you’re not alone in experiencing the impact the condition can cause” (5.7.PCA.BC-2).
		*Shared experience*	“Our poem appears to be a first opportunity for some of Eleven (as I now think of us) to experience their feelings in this way. I do not want it to stop. I KNOW those feelings, I have them EVERY day.....somehow there is comfort in sharing the awfulness of it all” (9.11.FTD.C-1). I enjoyed reading the poem because it was a thoughtful compilation of our shared experience with PCA. Hearing the thoughts and feelings of others living this challenge was comforting, encouraging, and empowering” (13.5.PCA.C-1); “I found reading it made me feel extremely emotional, it created a very strong response, seeing my words on the page amongst those of other people who are sharing different stages of this condition” (5.7.PCA.BC-2); “The completed poem makes me feel oddly secure. Caregiving is often a very isolating experience. Tying our reflections all together tethers our boats together” (10.5.PPA.C-3).
	Non-Resonating		“The poem does not meet the brief for me. As a means to inform those ignorant of FTD and its awfulness, it fails to elicit enough disgust, nor raise compassion towards those who daily battle it. I do thank the poet, of course” (9.11.FTD.C-2). “It was good to hear the words of others and to compare them (similarities and differences) to your own thoughts. We had thought that the ideas and feelings would be synthesized into a more holistic ‘statement’ that provided insight into the emotional and personal challenges experienced by this ‘group of people’ as a whole…(the poem) seems to be incomplete and lacks the emotional and insightful impact you would hope for in a poem of this nature” (13.10.PCA.C-2).

**Table 7 healthcare-12-00485-t007:** Themes and *sub-themes* from stakeholder group interviews.

Lived Experience and Reflection	*Hearing Own Experience in a different Way* *Validating Invisible Distress* *Multi-Voiced or Group Lived Experience* *Poetry Conveys Emotion and Meaning*
Curiosity and Exploration	*Creative aspect piquing interest* *Novel methodology and experience* *No wrong answers (gut response)*
Barriers and Inclusivity	*Am I a poetry person?* *Poetry as scary* *Reading all the poems was overwhelming*
Sharing the Poems	*Value as activity vs. value as product* *Where and how to disseminate* *Risks and benefits of sharing the poems*

**Table 8 healthcare-12-00485-t008:** Survey groups, number of participants and poems.

Participant Group	*N*	Poems
Group 1:		Hear My Pain
Professionals	14	Tumble Jumble
Students	16	Support
Group 2:		Eleven
Professionals	20	There’s So Much I Could Say
Students	13	Still Me
Group 3:		Steep Learning
Professionals	21	When Will I Know Longer Be Me
Students	9	Accept This Honour

## Data Availability

Data consisting of the completed poems, source material, and poet’s explanation of their creative process are available in Appendix A.

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
