# Peer review of "‘Misdiagnosed and Misunderstood’: Insights into Rarer Forms of Dementia through a Stepwise Approach to Co-Constructed Research Poetry"

_healthcare, 2024, doi:10.3390/healthcare12040485_

Round 1

Reviewer 1 Report

Comments and Suggestions for Authors

see attached file

Reviewer 2 Report

Comments and Suggestions for Authors

A minor point, but could 'rarer forms of dementia' be employed throughout the paper - rather than 'rarer dementia'. 'Rarer dementias' would also be fine.  'Rarer dementia' sounds like a discrete condition. (I recognise 'rare dementia support' is an established term.)

It is just a suggestion but maybe avoid the word 'typical' (line 47; line 67). This could imply Alzheimer's its experiential manifestations are routine or ordinary. 'More prevalent' or something similar could be suitable.

It would probably be picked up in the edit, but line 34 can state "poetry's capacity" (not "the poetry's capacity"). And it seems it should be 'programme', rather than 'programming' (line 66-67).

I think it would help the reader if the initialisms (e.g. diagnostic subtypes) were spelled out in full in table 1 and table 2). 

The aim(s) could perhaps be expressed more effectively (page 31). I feel the purpose of the paper is actually captured more effectively in the first line of the abstract. In this later section the lived experience is rather relegated to a supporting (ancillary) dimension. The delivery of the paper suggests the intention to grasp lived experience is an aim in its own right (and complementary to the intention to convey experience to healthcare professionals). A slight rewrite should achieve this balance. 

To boost transparency, the decision-making behind the choice of representative poems could be stated briefly (line 266).

I did wonder if this was almost two papers combined (it does cover a lot of ground and different participant groups/methods). However, the various elements are sufficiently well integrated. The findings, discussion and conclusion are lucidly and cogently conveyed. 

Round 2

Reviewer 1 Report

Comments and Suggestions for Authors

As a clinician, despite the effort demonstrated by the authors in the revision, I still find it challenging to read this article. I understand that the research is complex, but I believe that the paper should be overall comprehensible even for clinicians, especially if published in journals that recruit reviewers with a clinical background.
